# Logit Dispersion and the Scoring–Representation Gap in Medical OOD Detection

**Vinh Duc Tran**                                   VINH.TD225538@SIS.HUST.EDU.VN [1]

[1] *Hanoi University of Science and Technology (HUST), Vietnam*

## Abstract

Reliable medical triage requires uncertainty estimates that remain meaningful under domain shift. We study skin-lesion OOD detection under an acquisition shift from dermoscopic ISIC images to smartphone PAD-UFES-20 images. Our audit asks whether failure arises in the feature representation $f(x)$, the logit vector $z(x)$, or the scalar score derived from them.

We find a *Scoring–Representation Gap*: feature representations remain separable, but logit-level scores obscure this structure. Distribution-aware scoring on normalized features achieves AUROC = 0.953 and FPR95 = 0.296, whereas ELogitNorm logit-magnitude scoring yields AUROC = 0.742 and FPR95 = 0.812. This failure is linked to *Logit Dispersion*, defined as high relative variability of ID logit magnitudes, with $CV_{\mathrm{ID}} = \sigma(\|z(x)\|_2)/\mu(\|z(x)\|_2) = 1.11$. These results suggest that clinical OOD failures can arise from mismatched scoring assumptions, even when the representation remains informative.

**Keywords:** Uncertainty Quantification, Domain Shift, Logit Dispersion, Scoring–Representation Gap, Medical Triage.

## 1. Introduction

Deep learning models for skin-lesion triage must classify familiar cases accurately while assigning reliable uncertainty to images acquired under different clinical conditions. In skin imaging, such shifts arise when models trained on dermoscopic datasets are evaluated on smartphone photographs collected in less controlled settings. A central question is whether OOD failure reflects degraded representations, or whether useful OOD information remains present but is hidden by the final scoring rule.

We study this question through the hypothesis of a **Scoring–Representation Gap**: the feature space $f(x)$ may retain discriminative structure under shift, while the scalar score derived from logits $z(x)$ fails to expose it. In particular, we examine **Logit Dispersion**, where in-distribution (ID) logit magnitudes have high relative variability and overlap with shifted samples. This view complements prior work on feature collapse (Yang et al., 2025) by localizing failure to the interface between learned features and OOD scoring.

Our contributions are: (i) a compact clinical audit protocol that compares feature-, logit-, and score-space behavior under the ISIC-to-PAD-UFES-20 shift; (ii) an empirical characterization of Logit Dispersion as a failure mode for logit-magnitude scoring; and (iii) evidence that distribution-aware feature scoring can recover separability obscured at the logit level.

## 2. Methodology

### 2.1. Datasets and Clinical Shift

We evaluate OOD detection under an acquisition shift from dermoscopic to smartphone skin-lesion imaging. ISIC 2018 (Codella et al., 2019) serves as the ID source: its training split fits model and scoring parameters, and its validation split provides ID evaluation samples. PAD-UFES-20 (Pacheco et al., 2020) is used as the shifted OOD set. Compared with ISIC dermoscopy, PAD-UFES-20 contains consumer smartphone photographs acquired under less controlled conditions. The broad medical task is preserved, but acquisition process, image statistics, and nuisance factors change, making this a realistic covariate-shift testbed.

### 2.2. UQ Paradigms and Evaluation

All methods are evaluated as binary ID-vs-OOD detectors using ISIC validation images and PAD-UFES-20 samples. We report AUROC and FPR95, the false-positive rate on ID samples at 95% OOD sensitivity. Scores are oriented so that larger values indicate stronger OOD evidence.

We compare four scoring paradigms using a ResNet-18 backbone and verify the main trend on stronger architectures. MSP uses maximum softmax probability from a cross-entropy classifier. The contrastive baseline uses supervised contrastive learning (Khosla et al., 2020) with $k$-NN distance in feature space $f(x)$. The logit-based baseline follows ELogitNorm (Ding et al., 2025) and scores logit magnitude $\|z(x)\|_2$. Distribution-aware scoring applies Mahalanobis distance (Lee et al., 2018) to $\ell_2$-normalized features, with covariance estimated on ISIC training data using Ledoit-Wolf shrinkage (Ledoit and Wolf, 2004).

To quantify dispersion, we report the coefficient of variation of the ID score distribution, $CV_{\mathrm{ID}} = \sigma(s_{\mathrm{ID}})/\mu(s_{\mathrm{ID}})$, where $s_{\mathrm{ID}}$ is the scalar score over ID validation samples. For logit-magnitude scoring, $s(x) = \|z(x)\|_2$. Larger $CV_{\mathrm{ID}}$ indicates a more dispersed ID distribution, which can make threshold-based separation from shifted samples unreliable.

## 3. Results: Localizing the OOD Failure

Table 1 shows that the strongest separation occurs in feature space. Distribution-aware feature scoring achieves AUROC = 0.953 and FPR95 = 0.296, whereas logit-magnitude scoring reaches AUROC = 0.742 and FPR95 = 0.812. Thus, shifted samples remain detectable in $f(x)$, but the logit-level score fails to expose this signal.

The ablation in Table 1 further separates scoring from representation. Applying a distribution-aware metric directly to logits performs poorly, with AUROC = 0.319 and FPR95 = 0.998. Thus, logit space is not merely hard to score by magnitude; its geometry is poorly aligned with the shift, whereas normalized features support strong separation.

## 4. Discussion

The gap between feature-based and logit-based scoring supports a Scoring–Representation Gap under clinical acquisition shift. The representation $f(x)$ remains informative for separating ISIC from PAD-UFES-20, but this signal is weakened after projection into logit space

Table 1: OOD detection under the ISIC-to-PAD-UFES-20 acquisition shift. $CV_{\mathrm{ID}}$ is the coefficient of variation of the ID score distribution in the corresponding metric space.

| Paradigm | Metric Space | AUROC ↑ | FPR95 ↓ | $CV_{\mathrm{ID}}$ |
|---|---|---|---|---|
| MSP (Vanilla CE) | Softmax | $0.724 \pm 0.0001$ | $0.582 \pm 0.0001$ | — |
| Logit-based (ELogitNorm) | Logit $(z)$ | $0.742 \pm 0.0001$ | $0.812 \pm 0.0001$ | 1.11 |
| Contrastive | Feature $(f)$ | $0.767 \pm 0.009$ | $0.814 \pm 0.015$ | — |
| *Dist.-aware (Ablation)* | *Logit $(z)$* | *0.319* | *0.998* | *0.57* |
| Dist.-aware (LW) | Feature $(f)$ | $0.953 \pm 0.003$ | $0.296 \pm 0.004$ | 0.23 |

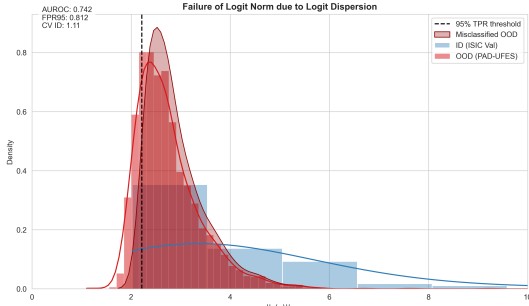 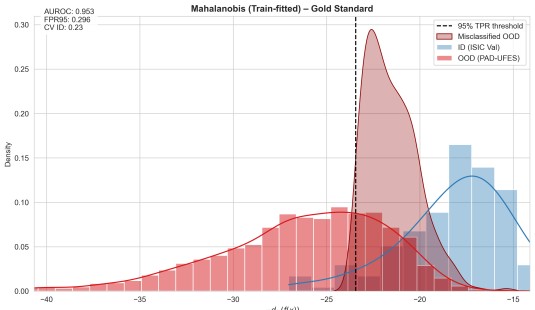

Figure 1: Score distributions under the ISIC-to-PAD-UFES-20 shift. Left: logit magnitude $\|z(x)\|_2$ shows **Logit Dispersion** ($CV_{\mathrm{ID}} = 1.11$), causing overlap with shifted samples and high FPR95. Right: Mahalanobis scoring on normalized features $f(x)$ yields a compact ID distribution ($CV_{\mathrm{ID}} = 0.23$) and clearer separation. Together, the panels illustrate the **Scoring–Representation Gap**: OOD signal is present in features but obscured by logit-level scoring.

and reduction to logit magnitude. The high $CV_{\mathrm{ID}}$ shows that ID logit magnitudes are already widely spread before OOD samples are considered, making threshold-based detection unreliable.

The logit-space Mahalanobis ablation localizes the failure further. Its poor performance suggests that the issue is not only the scalar magnitude score; the logit space itself appears geometrically distorted for this shift. In contrast, Mahalanobis scoring on normalized features, stabilized by Ledoit-Wolf covariance estimation (Ledoit and Wolf, 2004), preserves useful distributional structure. Medical OOD failures can therefore arise from a mismatch between feature geometry and scoring assumptions, rather than from complete representation collapse.

This short-paper audit focuses primarily on ResNet-18, with qualitative checks on stronger backbones. Broader validation across architectures, lesion datasets, foundation models, and scoring families is needed to determine how general Logit Dispersion is across medical imaging settings.

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
