# OpenReview forum: "Logit Dispersion and the Scoring–Representation Gap in Medical OOD Detection"
_MIDL.io/2026/Short_Papers — MIDL 2026 - Short Papers Poster_

### Official Review · Reviewer_r8kz · 2026-04-23
**Promising, but needs better presentation**

**Rating:** 2
**Confidence:** 4

**Review:**

Quality/Clarity: Poor. The paper could improve the description of some key terms (e.g., Logit Dispersion), the proposed audit framework, and its experimental design. Maybe a graphical abstract could summarize these and help save space.

Significance/Originality: Good. The results show promising preliminary results.

**Summary:**

The paper investigates failure detection under the ISIC-to-PADUFES20 distribution shift, arguing that common failure detection methods break down in this setting. The proposed explanation is that logit magnitudes remain similar across source and target distributions, undermining methods that rely on this signal.

**Strengths:**

- The investigation of domain shift between ISIC and PADUFES20 is relevant and potentially generalizable, as similar distributional mismatches are likely to arise in other dermatology datasets and beyond.
- The core hypothesis is worth pursuing. Figure 1 provides preliminary evidence that the proposed method can disentangle source and target distributions, which is an encouraging result.

**Weaknesses:**

- The method and experimental setup are not sufficiently described. Even within the tight page limit, a few concise sentences covering these aspects would substantially improve the paper's clarity and reproducibility. One practical way to recover space would be to convert the existing bullet-point lists to prose.
- The metric $CV_{ID}$ appears in the abstract but is never formally defined anywhere in the paper. A brief definition is necessary.

**Justification Of Rating:**

The paper addresses an interesting problem, and the preliminary findings are encouraging, but the contribution is not yet sufficiently developed to be presented at MIDL. I encourage the authors to expand the evaluation to additional domain shift scenarios and to use the available space more effectively to describe the method and experimental design.

---

### Decision · Program_Chairs · 2026-05-08

Accept (Poster)